# Adequate Segmentation in Marker-Based Motion Capture Studies for Hyperflexion and Hyperextension Lumbar Exercises

**DOI:** 10.3390/bioengineering12101087

**Published:** 2025-10-07

**Authors:** Claudia F. Romero-Flores, Rogelio Bustamante-Bello, Marcos Moya Bencomo, Iñaki Zenteno Aguirrezabal

**Affiliations:** 1Tecnologico de Monterrey, School of Engineering and Sciences, Ave. Eugenio Garza Sada 2501, Monterrey 64700, Nuevo Leon, Mexico; claudia.rf@tec.mx (C.F.R.-F.); marcosmoya@tec.mx (M.M.B.); 2Instituto de Ortopedia y Traumatología, Hospital Zambrano Hellion, TecSalud, San Pedro Garza García 66260, Nuevo Leon, Mexico; inaki@tecsalud.mx

**Keywords:** biomechanics, biomechanical characterization, lumbar vertebrae

## Abstract

The recent literature has debated the appropriate level of complexity for spine kinematic models. Multi-segmental analyses have been suggested to be more suitable for activities such as walking and running; however, studies focusing on sport-specific movements remain limited. This study compared four spine segmentation strategies for analyzing exercises simulating flexion and extension in acrobatic elements. Seventeen competitive university-level cheerleaders (male and female) participated in a motion capture study. Each athlete performed six exercises in the same order. Reflective markers were placed on the spinous processes of C7, T10, L1, L2, L3, L4, L5, and S1. From these, four models were constructed: (1) L1 and L5, (2) T10 and S1, (3) L1, L3, and L5, and (4) all lumbar vertebrae. Each model was fitted in the sagittal plane using a polynomial function and compared with the others via Pearson correlation. Model 3 (L1, L3, and L5) and Model 4 (all lumbar vertebrae) showed strong correlations across all trials, with Pearson coefficients approaching 1. These findings support the use of a two-segment representation of the lumbar spine (Model 3: L1–L3 and L3–L5) as a suitable approach for kinematic analysis of flexion–extension in acrobatic athletes.

## 1. Introduction

In acrobatic sports, athletes must develop back flexibility and improve their spine range of motion for the correct execution of several sports elements. Usually, these disciplines involve many spine flexion–extension repetitions that, combined with the high axial loading of the spine, are believed to contribute to the generation of back pain [1]. Additionally, when poor tumbling technique and excessive training are present, the risk of lumbar injuries increases [2]. Actually, acrobatic athletes have a high incidence of spine conditions such as back pain and scoliosis.

Spine injury prevalence in male and female artistic gymnastics is between 11.8% and 13% [3]. In rhythmic gymnastics, 19.6% of injuries are specifically located in the lower back [4]. Around 8–25% of professional and pre-professional dancers suffer from low back pain [5], and nearly 20% of the total injuries happen in the torso (with a high incidence in the lower back) [6]. Spine and head injuries were the principal cause of hospitalization between 2010 and 2019 in American cheerleaders as reported by the National Electronic Injury Surveillance System [7]. Overload spine injuries are the most prevalent injury type in professional ballet dancers and rhythmic gymnasts, which include mechanical low back pain, lumbar muscle injury, lumbar and cervical disc disease, neck muscle injury and low back facet syndrome [4,8].

Evidence supports that back pain and damaged musculoskeletal tissue alter standard motion patterns [9,10,11], which could result in poor technique performance. Studies on artistic gymnastics have shown that back muscles are less active during hyperflexion, and therefore, high loads are more likely to affect spine structures [12]. Additionally, risky motion patterns could damage tissue, such as ligaments, when left unattended. Because the spine is very complex, the spine biomechanics literature remains limited. Measuring the vertebral motion in vivo remains a challenge [13], resulting in few studies attempting to address the movement of the spine in sport-specific activities [14]. Therefore, there is a need in the scientific community to understand the relationship between spinal movement and injury risk factors [15]. To fully understand this relationship, it has been suggested that the spine motion study should look at each functional spinal unit (FSU) for a detailed understanding of its movement [15]. One FSU comprises two adjacent vertebrae and the intravertebral disc between them.

To measure FSU movement directly, researchers experiment with several models to provide valuable information about the maximum range of motion (ROM) between adjacent vertebrae. A review of rotation moments related to spinal structure stiffness measured in human cadavers can be found in [16]. Additionally, in vitro experiments have suggested that sagittal vertebral angles are not evenly distributed during flexion–extension exercises [17]. Animal models are also used due to anatomical similarities. Although these experiments provide essential and precise data on FSU movements, several limitations arise when applying this information to human in vivo scenarios: the limited number of specimen dispositions, the restricted variability in age and health of cadaveric tissues, the absence of muscle-driven motions, or the potential for inaccurate motion patterns. Hence, some critical information may be needed for specific human in vivo information [18].

Hence, in vivo studies are crucial to understanding the actual movement of vertebrae in real-life scenarios. However, several challenges are faced. The literature on spine hyperflexion and hyperextension kinematics is scarce, as most studies assess non-acrobatic populations with decreased ROM compared to acrobatic athletes. Although there are several spine ROM studies in the literature, they failed to address spine hypermobility in the sagittal plane [11]. Additionally, vertebral ROM capture is a challenge, and the researchers face two options: direct invasive measurements with imaging techniques or indirect non-invasive measurements with motion capture techniques. Although imaging techniques allow for accurate measurements, dynamic assessments are challenging and represent significant health risks to the participants due to prolonged ionic and radio-magnetic fields [19]. Consequently, motion capture techniques are used to measure spine movement safely, avoiding participant exposure to ionic fields. The maximum ROM of flexion and extension of the cervical spine has been studied by [20] and the thoracolumbar spine by [21,22].

However, because the spine is a complex structure comprising more than 100 joints (between facet joints, ribs, and intervertebral discs), it is necessary to define the optimal segmentation adequate for the activity to study. The scientific community has recently debated the optimal number of segments to divide the spine for study. In the early stages of biomechanics analysis, the torso (whole thoracolumbar spine) and the neck were considered rigid segments. Recent evidence suggests that more meaningful information is acquired when a multi-segmental study is carried out [5,23,24,25]. For torso studies, two or three adjacent segments have proven to have enough meaningful information [25]. However, scientists have not come to a consensus on the capture and processing protocols for spine biomechanics analysis [24]. Plug-in-Gait is a popular marker set when studying the lumbar spine [14], with the closer markers to the lumbar spine placed at T10 and sacrum. This research aims to compare different marker sets of the lumbar spine, from the traditional Plug-in-Gait, to identify the best segmentation for flexion–extension activities simulating acrobatic torso motions. Therefore, three models are proposed representing the lumbar spine as one segment, as two segments, and considering all five vertebrae.

## 2. Materials and Methods

### 2.1. Ethical Clearance

The experimental protocol was reviewed and accepted by the ethical committee ‘Colegio de Profesionales y Académicos de Medicina de Laboratorio’ (COPAMEL AC) in August 2023, under the name ‘Comparison of results obtained by protocol of optoreflective markers and inertial sensors for the capture of movement of the thoracolumbar spine in flexion and extension exercises in animation groups’, with the code 019.

### 2.2. Participants

After the recruitment process, a total of 20 university cheerleader athletes, both competitive squads, females and mixed, were recruited in the study (Table 1). Three participants were excluded from the present study, two due to extreme flexibility, and one due to a clinical history of vertebral fracture. Participants were included in the study if they had a minimum of 6 months of experience in the sport and were free of injuries. All athletes participated in a single hour-and-a-half capture session. Participants were asked to attend a single motion capture session in the Biomechanics laboratory of the institution. Participants were asked to wear tight shorts and leave their torsos as exposed as possible. Female participants were provided with a backless bespoke sports bra to leave the back skin free of clothes. The sports bra was secured with elastic therapeutic tape in specific areas where the participant needed extra support and comfort.

### 2.3. Data Capture

An eight-camera Vicon motion capture (Vicon Motion Systems Ltd., Oxford, UK) was used at 120 Hz to capture the movement of the participants during the exercises. Nine-millimetre spherical reflective markers were placed on specific body landmarks (Figure 1). The same researcher identified the vertebral spinous processes by palpation and marked the locations before marker placement. First, the C7 spinous process was identified by asking the participant to move their head to the sides and front. Next, the spinous processes of each vertebra between T1 and S1 were identified while the participant was prone. After completing vertebral location identification, reflective markers were placed on the landmarks with double-sided tape. Lateral lumbar markers were placed the same width as those at the left and right posterior superior iliac spine (PSIS). Additional markers in the pelvis were placed on both sides at the anterior superior iliac spine (ASIS) and the tip of the iliac crest (TIP). Requests to access the datasets should be directed to the corresponding author upon reasonable request.

### 2.4. Exercises

After marker placement, participants were guided through a warm-up that prepared them for the exercises and allowed them to familiarize themselves with the markers. Data capture was performed using Nexus 2.16 (Vicon Motion Systems Ltd., Oxford, UK). Participants completed four stationery and two dynamic exercises that were recorded twice, in the following order: static position (STA), standing trunk flexion and extension (FE, Figure 2a), sitting flexion (F, Figure 2b), cobra posture (COB, Figure 2c), shoulder flexion stretching (SHO, Figure 2d) and cobra posture with arms up (CUP, Figure 2e). Stationary exercises (STA, COB, SHO, CUP) were maintained for 20 s, and dynamic exercises (FE and F) were performed in five cycles during each recording. In the STA trial, the participant remained in a neutral standing position with feet separated at hip width and palms facing front. In the COB trial, participants were asked to start lying prone, raise their chests by strengthening both arms, hold that position for 20 s, and then return to the prone position. During the SHO trial, participants held a bar while kneeling and then pushed their chest towards the floor. Participants performed the cobra pose for the CUP trial but held a bar over their heads. In the standing FE exercise, participants were asked to fully flex and extend the torso without moving the pelvis. In the sitting F exercise, participants reached as far as possible to the front and returned to a neutral sitting position.

### 2.5. Data Processing

The best capture of each trial was selected for the analysis. Marker trajectories were filtered with a fourth-order low-pass Butterworth filter with a cut-off frequency of 5 Hz. Nexus 2.16 (Vicon Motion Systems Ltd., Oxford, UK) gap-filling tools were used to complete trajectories when they were missing. Lateral vertebral markers were used to reconstruct lumbar markers when needed. After gap filling was finished, all trajectories were exported into a csv file and processed in MATLAB R2023b (The MathWorks, Inc., Natick, MA, USA).

Lumbar marker trajectories were projected into the sagittal plane for analysis. The sagittal plane was defined as perpendicular to the line formed by the midpoints of PSIS and ASIS and parallel to the line formed by the midpoints of the right and left pelvis markers.

The torso (C7 to S1) angle relative to the pelvis was calculated to identify the beginning and end of stationary exercises and the maximum ROM for dynamic exercises, establishing the start and end of each cycle (Figure 3). Four cycles of the dynamic exercises were extracted, and time was normalized to achieve the cycle length. The analysis was performed in the directions of flexion (from max extension to max flexion) and extension (from max flexion to max extension). Each cycle was analyzed at 0, 25, 50, 75 and 100% of the cycle’s full length. Four captures distributed during the trial duration were extracted for stationary exercises. A total of 68 captures were processed for each stationary exercise and each cycle percentage in dynamic exercises.

Four marker positions were selected to represent the lumbar spine: Figure 4a model 1 lumbar as a whole segment (L1 and L5), Figure 4b model 2 using the so common Plug-in-Gait markers close to the lumbar spine (T10 and S1), Figure 4c model 3 lumbar as two equidistant segments (L1, L3 and L5), and Figure 4d model 4 representing all FSU of the lumbar spine (one marker in each vertebra from L1 to L5). Figure 4 illustrates the markers used in the experiment and for analyzing each model.

Each model was fitted with a polynomial line, using all model markers, to represent the lumbar spine. Models 1 and 2 used a first-degree polynomial because only two markers were available, while models 3 and 4 used a second-degree polynomial. The longitudinal axis was selected as the independent variable. The lumbar lordosis was calculated for each analyzed position using the circle-fit method [26]. This method represents the curvature as the angle between the coincident radius of L1 and L5. Relative lumbar lordosis was calculated as the difference between the analyzed position and the STA trial.

### 2.6. Statistical Analysis

Each model polynomial fit was evaluated at five equidistant points of the independent variable and compared to the other model lines to calculate the correlation between models. Pearson correlation between models was calculated using the corr function in MATLAB 2023b (The MathWorks, Inc.). As stated before, 68 values were used to calculate the correlation between models for each exercise.

## 3. Results

Pearson correlation values between models are presented in Table 2. Models 3 and 4 show high correlation values in all cases, with standard deviation values smaller than 0.05, except for 0% flexion in FE and 100% extension in FE (0.09 and 0.09, respectively). Models 1 and 3 showed strong correlations with F exercise throughout the movement in both flexion and extension directions. The same happened with models 1 and 4. Lumbar lordosis showed a high variation between participants in all cases.

## 4. Discussion

This work compared several reflective marker positions to understand the motion of the lumbar spine. Model 4 (Figure 1) was studied because it offers the most realistic option for researchers addressing vertebral motion using marker-based motion capture, as it considers all lumbar vertebrae. Models 1, 2 and 3 (Figure 1) were chosen because they represent the widely used marker sets in the research industry. During this experiment, which attempted to achieve hyperflexion and hyperextension in simulated acrobatic torso positions, models 3 and 4 (Figure 1) performed reasonably similarly. Overall, the Pearson correlation between these two models was 1 (Table 2), except for FE maximum extension (0% in flexion and 100% in extension). Yet, the correlation between models 3 and 4 always stayed above 0.99. Standard deviation values are also relatively small (less than 0.09), indicating that both models had a strong correlation among the participants. The correlation between Model 4 and the other two models (1 and 2) resulted in Pearson values between 0.66 and 0.99, and standard deviation values between 0.06 and 0.55. This result suggests that when addressing maximum flexion and extension in acrobatic athletes, two segments (Model 3) to represent the lumbar spine give the same information as tracking all vertebrae (Model 4). A reason for these results might be found in the range of motion between adjacent vertebrae. The flexion–extension between the lumbar FSU is relatively small, between 10 and 14 degrees [27]. Additionally, some studies have attempted to describe the curvature of the spine using a spline fitting between skin markers and magnetic resonance imaging. The RMSE errors have been found to be smaller than 11 ± 3.7 mm [28]. Leardini et al. [29] proposed a 5-link-segment model to represent the thoracolumbar spine, which defined the lumbar spine as two segments using the same markers as Model 3. The same lumbar marker position as Model 3, revealed significant differences in spine angles while studying common dance elements [30]. They reported a good balance between segmentation and feasibility in locomotion and elementary exercises. On the other hand, ref. [25] concluded that two or three liked segments were the best when representing the thoracolumbar spine in walking trials. It is reasonable that fewer segments are needed when the torso shape changes are minimal (such as walking or running) but fail when representing more extensive shape changes in the torso (such as in maximum flexion or extension). A suitable resolution is needed to avoid under- or overestimating calculated results. Nevertheless, the results of the present study support two segments for the lumbar spine in whole spine flexion and extension. For instance, with a similar approach, in male gymnastics floor exercises, maximum flexion and extension lordosis angles have been reported up to 65° and 12° at landing, respectively, where ground reaction forces can be up to 10 body weights [12].

Looking closely at FE and F movement directions, one could think that going from maximum flexion to maximum extension would be the reverse of maximum extension to maximum flexion. However, it seems that participants exhibit different, albeit similar, movement patterns, as evidenced by the varying correlation values between corresponding percentages (0% flexion with 100% extension, 25% flexion with 75% extension, and so on). This suggests that muscle activation may influence vertebral movement patterns, a finding that warrants further investigation. Differences in spine kinematics in dancers have been shown to depend on the motion direction [30]. Young and elderly individuals have also shown different motion patterns between flexion and extension in thoracic segments [31].

Although this study addressed the motion of acrobatic athletes, the authors must mention that the current participants were not among the most flexible athletes. A higher rate of back flexibility might change the results found during this work. Two participants had to be removed from the analysis due to their flexibility in extension, which occluded markers placed on the back. So, the camera set, and marker positioning are critical when measuring extremely flexible athletes. Therefore, caution is recommended when extrapolating the results presented in this study to extremely flexible people. Moreover, other marker placements might be needed for extreme back flexibility. A limitation of this study is the number of participants acquired. Only 17 athletes participated in the experiment. To cope with the limited population, four samples for each trial were analyzed. A larger sample size could help to characterize the back flexibility of acrobatic athletes. In this study, the relative lumbar lordosis angle was used to compare the back flexibility of participants because the circle fit method is better for understanding changes in curvature rather than the absolute values [26]. Curvature changes greater than 57 and −18 degrees in flexion and extension might behave differently between Models 3 and 4.

Caution is advised when concluding any marker-based motion capture study, including this research. It is known that soft tissue artefacts affect marker placement and movement. However, back marker errors stay small for normal BMI ranges during seating [32]. Still, some authors advise researchers to keep in mind that soft tissue artefacts have a non-uniform pattern on the spine, which can significantly alter the errors in the measurement (between 17° and 21° when measuring curvature) [26]. For the lumbar vertebrae, soft tissue artefacts have been reported to be as large as 14.8 mm [33]. However, the skin marker kinematic measurements are still reliable as the correlation between bone and skin markers is linear [33]. Some anatomical locations may create bigger errors. It has been reported that for the torso inclination angle, reflective markers located at T3 and T7 are recommended [34].

Future work should focus on the contribution of each FSU to the total lumbar angle to further understand spine motion and flexibility and address kinetic parameters to understand loading patterns and their relationship with workloads and tissue damage.

## 5. Conclusions

In conclusion, this work analyzed the correlation between four lumbar marker placements during flexion–extension activities simulating acrobatic positions. Model 3 (markers at L1, L3 and L5) has a good equilibrium between simplicity and resolution when analyzing full lumbar flexion extension of acrobatic athletes in the sagittal plane. Given its strong correlation with Model 4, which includes all five vertebrae and provides the most comprehensive information available in a marker-based study, it is considered the most reliable. Model 3 performs well in acrobatic athletes with greater back flexibility than the general population. However, it has not been tested in highly flexible athletes.

## Figures and Tables

**Figure 1 bioengineering-12-01087-f001:**
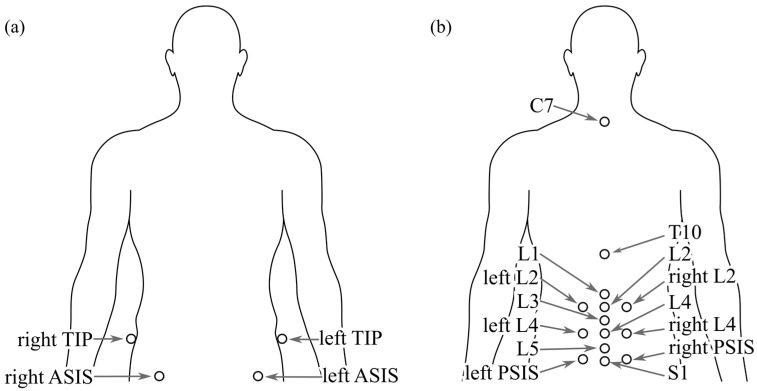
(**a**) Front and (**b**) back view of the markers used during the experiment. Markers were placed at the spinous processes of C7, T10, L1, L2, L3, L4, L5 and S1, laterally of L2 and L4, left and right of the posterior superior iliac spine (PSIS), the anterior superior iliac spine (ASIS) and the tip of the iliac crest (TIP).

**Figure 2 bioengineering-12-01087-f002:**
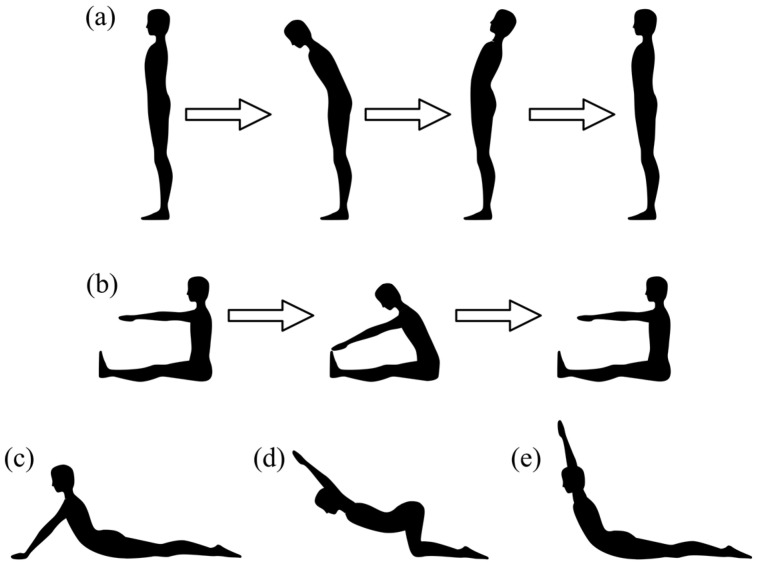
Exercise batch performed by the athletes, (**a**) standing FE, (**b**) sitting F, (**c**) COB, (**d**) SHO and (**e**) CUP.

**Figure 3 bioengineering-12-01087-f003:**
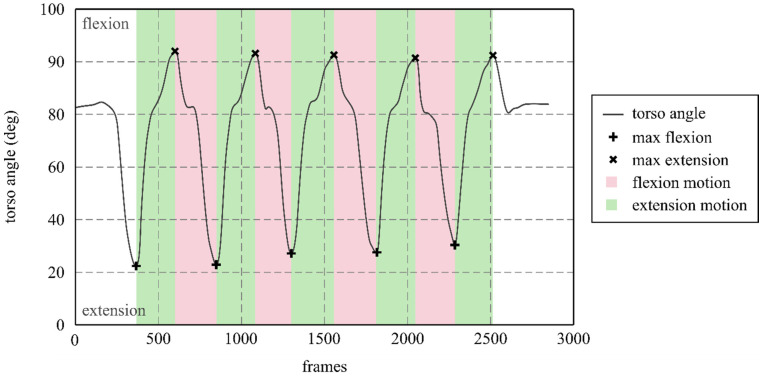
Event and cycle identification using the torso angles. The minimum value represents the maximum flexion (+), while the maximum value represents the maximum extension (x). Flexion motion (pink shade) goes from maximum extension to maximum flexion, while extension motion (green shade) goes from maximum flexion to maximum extension.

**Figure 4 bioengineering-12-01087-f004:**
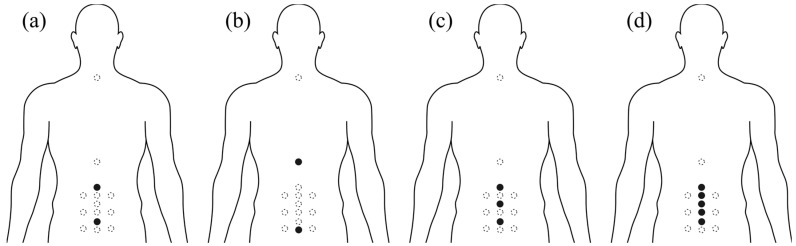
Marker placement and selection for each model. Markers selected for each model are represented in solid black, while unused markers for the specific model are shown in a dashed grey line.

**Table 1 bioengineering-12-01087-t001:** Participant characteristic summary.

Gender (n)	Age (years)	Height (cm)	Weight (kg)	BMI
males (6)	19.67 ± 1.11	178.47 ± 8.08	75.02 ± 13.81	23.45 ± 3.28
females (11)	20.18 ± 1.8	160.27 ± 6.58	53.92 ± 18.84	20.93 ± 7.32

**Table 2 bioengineering-12-01087-t002:** Mean and standard deviation values for Pearson correlation between models and lumbar lordosis curvature angle for each exercise. Pearson correlation values higher than 0.95 are shown in bold characters, and standard deviations smaller than 0.05 are shown with light grey shading.

	Model Comparison						
	1 vs. 2	1 vs. 3	1 vs. 4	2 vs. 3	2 vs. 4	3 vs. 4	Relative Lumbar Lordosis (deg)
STA							
	0.76 ± 0.65	0.74 ± 0.31	0.76 ± 0.30	0.69 ± 0.42	0.66 ± 0.48	**1.00 ± 0.01**	
FE in flexion							
0%	**1.00 ± 0.00**	0.78 ± 0.25	0.75 ± 0.33	0.78 ± 0.25	0.75 ± 0.33	**0.99 ± 0.09**	−13.74 ± 11.75
25%	0.94 ± 0.34	0.78 ± 0.26	0.77 ± 0.28	0.76 ± 0.31	0.75 ± 0.34	**1.00 ± 0.01**	−3.34 ± 8.61
50%	**1.00 ± 0.00**	0.82 ± 0.22	0.81 ± 0.25	0.82 ± 0.22	0.81 ± 0.25	**1.00 ± 0.01**	3.29 ± 8.91
75%	0.94 ± 0.34	0.93 ± 0.11	0.92 ± 0.12	0.87 ± 0.33	0.87 ± 0.32	**1.00 ± 0.01**	20.84 ± 19.05
100%	0.91 ± 0.41	**0.96 ± 0.12**	**0.96 ± 0.11**	0.89 ± 0.38	0.89 ± 0.39	**1.00 ± 0.00**	31.98 ± 26.82
FE in extension							
0%	0.94 ± 0.34	**0.97 ± 0.10**	**0.97 ± 0.10**	0.91 ± 0.35	0.91 ± 0.35	**1.00 ± 0.00**	32.31 ± 27.18
25%	0.91 ± 0.41	0.87 ± 0.19	0.88 ± 0.17	0.83 ± 0.32	0.83 ± 0.33	**1.00 ± 0.00**	15.40 ± 14.73
50%	**1.00 ± 0.00**	0.87 ± 0.14	0.87 ± 0.15	0.87 ± 0.14	0.87 ± 0.15	**1.00 ± 0.00**	1.47 ± 11.37
75%	**1.00 ± 0.00**	0.80 ± 0.22	0.79 ± 0.23	0.80 ± 0.22	0.79 ± 0.23	**1.00 ± 0.01**	−7.32 ± 10.66
100%	**1.00 ± 0.00**	0.78 ± 0.25	0.75 ± 0.33	0.78 ± 0.25	0.75 ± 0.33	**0.99 ± 0.09**	−13.65 ± 11.80
F in flexion							
0%	0.88 ± 0.47	**0.97 ± 0.12**	**0.97 ± 0.16**	0.90 ± 0.39	0.91 ± 0.38	**1.00 ± 0.01**	34.94 ± 23.10
25%	**1.00 ± 0.00**	**0.99 ± 0.02**	**0.99 ± 0.01**	**0.99 ± 0.02**	**0.99 ± 0.01**	**1.00 ± 0.00**	41.40 ± 18.48
50%	0.94 ± 0.34	**0.97 ± 0.10**	**0.97 ± 0.12**	0.94 ± 0.25	0.94 ± 0.24	**1.00 ± 0.00**	50.30 ± 16.77
75%	0.88 ± 0.47	**0.96 ± 0.12**	**0.95 ± 0.15**	0.89 ± 0.38	0.89 ± 0.36	**1.00 ± 0.00**	56.42 ± 15.69
100%	0.76 ± 0.65	**0.96 ± 0.09**	**0.95 ± 0.11**	0.78 ± 0.57	0.79 ± 0.55	**1.00 ± 0.00**	57.30 ± 16.51
F in extension							
0%	0.79 ± 0.61	**0.96 ± 0.08**	**0.96 ± 0.10**	0.81 ± 0.53	0.81 ± 0.51	**1.00 ± 0.00**	57.16 ± 16.63
25%	0.82 ± 0.57	**0.96 ± 0.10**	**0.95 ± 0.12**	0.84 ± 0.48	0.84 ± 0.46	**1.00 ± 0.00**	55.88 ± 16.04
50%	**0.97 ± 0.24**	0.93 ± 0.19	0.94 ± 0.17	0.92 ± 0.22	0.94 ± 0.18	**1.00 ± 0.02**	47.50 ± 17.06
75%	0.88 ± 0.47	**0.97 ± 0.11**	**0.97 ± 0.14**	0.89 ± 0.41	0.90 ± 0.40	**1.00 ± 0.01**	39.29 ± 15.67
100%	0.88 ± 0.47	**0.97 ± 0.12**	**0.97 ± 0.16**	0.90 ± 0.39	0.91 ± 0.38	**1.00 ± 0.01**	34.95 ± 23.10
COB							
	**1.00 ± 0.00**	0.90 ± 0.15	0.90 ± 0.16	0.90 ± 0.15	0.90 ± 0.16	**1.00 ± 0.00**	−17.91 ± 17.87
SHO							
	0.91 ± 0.41	0.83 ± 0.25	0.83 ± 0.23	0.79 ± 0.34	0.79 ± 0.34	**1.00 ± 0.01**	−14.98 ± 20.68
CUP							
	0.94 ± 0.34	**0.95 ± 0.06**	0.94 ± 0.06	0.89 ± 0.33	0.90 ± 0.31	**1.00 ± 0.01**	−18.30 ± 28.91

## Data Availability

The datasets presented in this article are not readily available because it is part of an ongoing project, and confidentiality protects data from being publicly available until the end of the project. Requests to access the datasets should be directed to the corresponding author upon reasonable request.

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
