# Peer review of "Adequate Segmentation in Marker-Based Motion Capture Studies for Hyperflexion and Hyperextension Lumbar Exercises"

_bioengineering, 2025, doi:10.3390/bioengineering12101087_

Round 1
Reviewer 1 Report
Comments and Suggestions for Authors
The reviewer understood the paper proposed the appropriate segmentation of lumber spine to represent acrobatic motion based on experimental data analysis. The research purpose and their background were clearly mentioned based on literatures. The method is appropriate and well described in the paper. I have some editorial comments to improve the quality of the paper:
(1) hyphenation
Some unnecessary hyphenation is still remaining in the line (not end of the line).
For example,
2.2. Participants line-123: par-ticipant
2.2. Participants line-125: mark-ers
2.4. Exercises line-138: sec-onds
2.4. Exercises line-139: re-cording
2.4. Exercises line-147: pos-sible
(2) Abstract
English quality of the main text is fine. Whereas, the abstract should be improved. I suggest as follows:
Recent literature has debated the appropriate level of complexity for spine kinematic models. Multi-segmental analyses have been suggested to be more suitable for activities such as walking and running; however, studies focusing on sport-specific movements remain limited. This study compared four spine segmentation strategies for analyzing exercises simulating flexion and extension in acrobatic elements. Seventeen competitive university-level cheerleaders (male and female) participated in a motion capture study. Each athlete performed six exercises in the same order. Reflective markers were placed on the spinous processes of C7, T10, L1, L2, L3, L4, L5, and S1. From these, four models were constructed: (1) L1 and L5, (2) T10 and S1, (3) L1, L3, and L5, and (4) all lumbar vertebrae. Each model was fitted in the sagittal plane using a polynomial function and compared with the others via Pearson correlation. Model 3 (L1, L3, and L5) and Model 4 (all lumbar vertebrae) showed strong correlations across all trials, with Pearson coefficients approaching 1. These findings support the use of a two-segment representation of the lumbar spine (Model 3: L1–L3 and L3–L5) as a suitable approach for kinematic analysis of flexion–extension in acrobatic athletes.
Author Response
Comments 1: hyphenation
Some unnecessary hyphenation is still remaining in the line (not end of the line).
For example,
2.2. Participants line-123: par-ticipant
2.2. Participants line-125: mark-ers
2.4. Exercises line-138: sec-onds
2.4. Exercises line-139: re-cording
2.4. Exercises line-147: pos-sible
Response 1: Thank you for pointing this out. We agree with this comment. Therefore, we have carefully checked the document and erased all in-text hyphenation. Words in lines 132, 134, 141, 148 and 157 were corrected.
Comments 2: Abstract
English quality of the main text is fine. Whereas, the abstract should be improved. I suggest as follows:
Recent literature has debated the appropriate level of complexity for spine kinematic models. Multi-segmental analyses have been suggested to be more suitable for activities such as walking and running; however, studies focusing on sport-specific movements remain limited. This study compared four spine segmentation strategies for analyzing exercises simulating flexion and extension in acrobatic elements. Seventeen competitive university-level cheerleaders (male and female) participated in a motion capture study. Each athlete performed six exercises in the same order. Reflective markers were placed on the spinous processes of C7, T10, L1, L2, L3, L4, L5, and S1. From these, four models were constructed: (1) L1 and L5, (2) T10 and S1, (3) L1, L3, and L5, and (4) all lumbar vertebrae. Each model was fitted in the sagittal plane using a polynomial function and compared with the others via Pearson correlation. Model 3 (L1, L3, and L5) and Model 4 (all lumbar vertebrae) showed strong correlations across all trials, with Pearson coefficients approaching 1. These findings support the use of a two-segment representation of the lumbar spine (Model 3: L1–L3 and L3–L5) as a suitable approach for kinematic analysis of flexion–extension in acrobatic athletes.
Response 2: Thank you for pointing this out. We agree with the version you have kindly provided of the abstract. We have made the changes to match the version currently offered.
Reviewer 2 Report
Comments and Suggestions for Authors
This manuscript compareed four different spine segmentation models for analysing flexion and extension in university cheerleaders. Revisions are needed to improve the quality and clarity of the manuscript as follows:
- Title: Authors should consider making it clearer and more pertinent to the content of the study i.e, What is Adequate resolution? The participant is cheerleaders, not general population, and also not an athlete.
- The rationale and background information of the four marker models selected for this study, and also other previously studied models, should be clearly provided in the Introduction section, along with previous studies using any of these models in terms of accuracy and limitations.
- The EC COA number should also be stated. In addition, the English title of the approved protocol if available.
- It is not clear to me about this statement "Using a significance level of 5%, a probability of 50%, a standard deviation of 15° and a margin of 5°, a 49-sample size was calculated. After the recruitment process, a total of 17 university cheerleader athletes." From my understanding, the calculated sample size was 49, Why were only 17 participants recruited? Would this be sufficient for statistical analysis based on the calculated sample size?
- Statement "Fifty-three nine-millimetre spherical reflective markers were placed on specific body landmarks (Figure 1)." I could not see 53 markers in figure 1 as stated.
- Please recheck the use of unnecessary hyphens in several sentences i.e while the par-ticipant was prone. After completing vertebral location identification, reflective markers were placed on the landmarks with double-sided tape. Lateral lumbar mark-ers./ participants reached as far as pos-sible to the front and returned to a neutral sitting position.
- Please state the name of the software, version and manufacturer used for motion analysis.
- A section of statistical analysis should be added.
- Please further discuss why model 3 (two equidistant segments) is correlated well with model 4 (FSU of the lumbar spine), not model 1 (lumbar as a whole segment) or model 2 ( a common Plug-in-Gait markers).
English should be checked for writing mistakes and also to improve comprehension.
Author Response
Comments 1: Title: Authors should consider making it clearer and more pertinent to the content of the study i.e, What is Adequate resolution? The participant is cheerleaders, not general population, and also not an athlete.
Response 1: Thank you for pointing this out. We agree with this comment and acknowledge that the word ‘resolution’ might create confusion for the reader. Therefore, we have referred to it as ‘segmentation’ and changed the word throughout the document (including the title).
Regarding the participants, cheerleaders were selected because they fall into the category of acrobatic athletes, as they performed acrobatic movements during their routines and training. We have added some information regarding cheerleading spine injuries in lines 44-46.
Comments 2: The rationale and background information of the four marker models selected for this study, and also other previously studied models, should be clearly provided in the Introduction section, along with previous studies using any of these models in terms of accuracy and limitations.
Response 2: Thank you for pointing this out. We agree with this comment. Therefore, we have added some information regarding the marker set used in the study. You can find that information in lines 95-96, 98-99, and 102-103.
Comments 3: The EC COA number should also be stated. In addition, the English title of the approved protocol if available.
Response 3: Thank you for pointing this out. We agree with this comment. Unfortunately, an ethical acceptance number was not provided. We have discussed this with the editorial board, and they have asked for additional information. We have provided the English translation title instead of the Spanish version in the manuscript (lines 109-111)
Comments 4: It is not clear to me about this statement "Using a significance level of 5%, a probability of 50%, a standard deviation of 15° and a margin of 5°, a 49-sample size was calculated. After the recruitment process, a total of 17 university cheerleader athletes." From my understanding, the calculated sample size was 49, Why were only 17 participants recruited? Would this be sufficient for statistical analysis based on the calculated sample size?
Response 4: Thank you for pointing this out. We agree with this comment. Therefore, we have omitted the sample calculation to make it more straightforward. As that calculation was intended for another study using the same participants, it creates confusion for the reader. Additionally, we have added a limitation in the discussion, addressing the number of samples for the statistical analysis (lines 265-267).
Comments 5: Statement "Fifty-three nine-millimetre spherical reflective markers were placed on specific body landmarks (Figure 1)." I could not see 53 markers in figure 1 as stated.
Response 5: Thank you for pointing this out. We agree with this comment. The 53 markers were used for other studies and calculations. To avoid reader confusion, we have only referred to the markers used during the processing of the current manuscript (omitting Fifty-three in line 127). Therefore, we have updated Figure 1 so that anterior and posterior views are shown. We updated the figure caption accordingly. Additionally, we have edited Figure 4 to show the markers used for the current study clearly and highlighted the markers used in each model.
Comments 6: Please recheck the use of unnecessary hyphens in several sentences i.e while the par-ticipant was prone. After completing vertebral location identification, reflective markers were placed on the landmarks with double-sided tape. Lateral lumbar mark-ers./ participants reached as far as pos-sible to the front and returned to a neutral sitting position.
Response 6: Thank you for pointing this out. We agree with this comment. Therefore, we have carefully checked the document and erased all in-text hyphenation. Words in lines 132, 134, 141, 148 and 157 were corrected.
Comments 7: Please state the name of the software, version and manufacturer used for motion analysis.
Response 7: Thank you for pointing this out. We agree with this comment. Therefore, we have added the requested information for data processing in lines 165 and 166.
Comments 8: A section of statistical analysis should be added.
Response 8: Thank you for pointing this out. We agree with this comment. Therefore, we have added in line 204 the information related to the statistical test used for the results.
Comments 9: Please further discuss why model 3 (two equidistant segments) is correlated well with model 4 (FSU of the lumbar spine), not model 1 (lumbar as a whole segment) or model 2 ( a common Plug-in-Gait markers).
Response 9: Thank you for pointing this out. We agree with this comment. Therefore, we have added some additional sentences explaining this conclusion (lines 220, 226-230, 235-236)
Round 2
Reviewer 1 Report
Comments and Suggestions for Authors
No comments.
Author Response
Response: Thank you for providing your time for the revision of the manuscript. We appreciate the effort and time taken to read the manuscript. And believe your contribution made the manuscript stronger.
Reviewer 2 Report
Comments and Suggestions for Authors
Authors have revised the manuscript, but some points are still unclear as follow:
- COA number, I believe that all EC will issue a protocol approval number or COA number. The reply from the authors that EC did not issue a COA number is rather odd.
- Sample size calculation, authors reply that the calculation was intended for another study. How did authors get the sample size of 17 for this study. Please give the calculation or reason for such number with references.
- Concerning the statistical analysis, I suggested writing a subsection on how to perform, what software was used? version? Manufacturer? parameters? , not just one line as presented.
- Discussion : I suggested to further discuss why model 3 was closed to model 4, not others. I mean to discuss the possible reason behind the good or bad correlation for each model with cited references, not just the further explanation of the results that the correlation values were these numbers.
Author Response
Comments 1: COA number, I believe that all EC will issue a protocol approval number or COA number. The reply from the authors that EC did not issue a COA number is rather odd.
Response 1: We are in touch with the ethical committee and are working on a solution for this situation.
Comments 2: Sample size calculation, authors reply that the calculation was intended for another study. How did authors get the sample size of 17 for this study. Please give the calculation or reason for such number with references.
Response 2: Thank you for pointing this out. The sample size calculation was for a bigger study on spine motion. For the present study, we were able to recruit 20 participants within the study's time span, but could only include 17 in the manuscript. Two presented extreme flexibility, occluding the markers with their own body, making it impossible to do our calculations. The third reported having a compression lumbar fracture one year before the experiment was done. We have added this information in lines 112-115. We conducted the sample size calculation; however, we were unable to recruit the necessary number of participants for a strong statistical representation. We have acknowledged it in the discussion section (lines 270-272)
Comments 3: Concerning the statistical analysis, I suggested writing a subsection on how to perform, what software was used? version? Manufacturer? parameters? , not just one line as presented.
Response 3: Thank you for pointing this out. We agree with this comment. Therefore, we have added a Statistical section explaining how the Pearson value was obtained (lines 205-209). We apologise for misunderstanding the round 1 comment concerning this topic.
Comments 4: Discussion : I suggested to further discuss why model 3 was closed to model 4, not others. I mean to discuss the possible reason behind the good or bad correlation for each model with cited references, not just the further explanation of the results that the correlation values were these numbers.
Response 4: Thank you for pointing this out. However, we find it challenging to provide cited references to this particular section of the discussion, as it is the main finding of the manuscript. To the best of our knowledge, there is no other study comparing the correlation between different spine marker sets.
Round 3
Reviewer 2 Report
Comments and Suggestions for Authors
Authors have replied and revised accordingly. However, two points remained to be questioned.
- The ethical concern about the approval of this study. I will let the editorial office to handle this matter and make a decision.
- The discussion about the possible reason behind the good or bad correlation among the models employed. Although the authors replied that they could not find any cited references since this study was the first of its kind. To me, no study now was the first in the world. The knowledge nowadays is incremental, and the results should be discussed based on scientific reasons with available prior cited references to support such discussion. Not just a results explanation in a Discussion section.
Author Response
Comments 1: The ethical concern about the approval of this study. I will let the editorial office to handle this matter and make a decision.
Response 1: Thank you for procuring the ethical integrity of the study. We have finally got in touch with the ethical committee, and have provided us with the approval number for the experiment, as well as the updated approval letter. The information regarding the approval code can be found on line 110.
Comments 2: The discussion about the possible reason behind the good or bad correlation among the models employed. Although the authors replied that they could not find any cited references since this study was the first of its kind. To me, no study now was the first in the world. The knowledge nowadays is incremental, and the results should be discussed based on scientific reasons with available prior cited references to support such discussion. Not just a results explanation in a Discussion section.
Response 2: Thank you for pointing this out. We have provided some additional lines in an effort to explain the possible reasons behind the results (lines 235 -239). Additionally, lines from 239 to 248 were also written in an attempt to explain the reasoning behind the results.